# A New SNP-Based Genotyping Method for *C. psittaci*: Application to Field Samples for Quick Identification

**DOI:** 10.3390/microorganisms9030625

**Published:** 2021-03-17

**Authors:** Fabien Vorimore, Rachid Aaziz, Bertille de Barbeyrac, Olivia Peuchant, Monika Szymańska-Czerwińska, Björn Herrmann, Christiane Schnee, Karine Laroucau

**Affiliations:** 1Laboratory for Animal Health, Bacterial Zoonosis Unit, ANSES Maisons-Alfort, Paris-Est University, 94706 Paris, France; Fabien.vorimore@anses.fr (F.V.); rachid.aaziz@anses.fr (R.A.); 2Mycoplasma and Chlamydia Infections in Humans, University of Bordeaux, 33076 Bordeaux, France; bertille.de-barbeyrac@u-bordeaux.fr (B.d.B.); olivia.peuchant@u-bordeaux.fr (O.P.); 3Department of Cattle and Sheep Diseases, National Veterinary Research Institute, 24100 Pulawy, Poland; monika.szymanska@piwet.pulawy.pl; 4Department of Clinical Microbiology, Uppsala University Hospital, 75185 Uppsala, Sweden; bjorn.herrmann@medsci.uu.se; 5Section of Clinical Bacteriology, Department of Medical Sciences, Uppsala University, 75123 Uppsala, Sweden; 6Institute of Molecular Pathogenesis, Friedrich-Loeffler-Institut (Federal Research Institute for Animal Health), 07743 Jena, Germany; Christiane.Schnee@fli.de

**Keywords:** *Chlamydia psittaci*, SNP, PCR-high-resolution melting (HRM), psittacosis, avian chlamydiosis, genotyping

## Abstract

*Chlamydia* (*C*.) *psittaci* is the causative agent of avian chlamydiosis and human psittacosis. In this study, we extracted single-nucleotide polymorphisms (SNPs) from the whole genome sequences of 55 *C. psittaci* strains and identified eight major lineages, most of which are host-related. A combined PCR/high-resolution melting (HRM) assay was developed to screen for eight phylogenetically informative SNPs related to the identified *C. psittaci* lineages. The PCR-HRM method was validated on 11 available reference strains and with a set of 118 field isolates. Overall, PCR-HRM clustering was consistent with previous genotyping data obtained by *omp*A and/or MLST analysis. The method was then applied to 28 *C. psittaci*-positive samples from animal or human cases. As expected, PCR-HRM typing results from human samples identified genotypes linked to ducks and pigeons, a common source of human exposure, but also to the poorly described Mat116-like genotype. The new genotyping method does not require time-consuming sequencing and allows a quick identification of the source of infection.

## 1. Introduction

*Chlamydia* (*C*.) *psittaci*, an agent from the *Chlamydiaceae* family, is commonly isolated from a wide range of birds worldwide [1]. This species is typically associated with infection in humans in close contact with birds, following inhalation of aerosolized infectious particles originating from dry feces and respiratory secretions. Pet bird owners and breeders, pet shop and zoo employees, poultry workers, veterinarians, laboratory technicians, and wildlife workers are particularly at risk [2]. *C. psittaci*-infected humans usually exhibit non-specific signs, including fever, headache, myalgia and non-productive cough. Misdiagnosis and/or inappropriate antibiotic-based treatment can also result in death due to atypical pneumonia. Avian species belonging to the orders Psittaciformes, Galliformes, Anseriformes, and Columbiformes are common sources of infection [3]. In birds, the clinical presentation may vary considerably and is influenced by the pathogen (genotypes of *C. psittaci*) and the host (species, age, health and immunological status). Alongside subclinical infections, severe respiratory, digestive and ocular forms are described [2].

The different *C. psittaci* typing methods developed over time have revealed a strain diversity within this species and a close association between serotypes/genotypes and bird groups. The initially described eight serovars (A to F for avian strains, WC and M56 for mammalian strains), based on the use of monoclonal antibodies [4,5,6] were found to be equivalent with *omp*A-based genotypes. Later, the *omp*A sequence analysis from a large panel of isolates led to the introduction of new *C. psittaci* provisional genotypes (1V, 6N, Mat116, R54, YP84 and CPX0308) [7]. Recently, 1V strain was re-affiliated to the *C. abortus* species [8], based on MLST analysis of seven conserved household genes [9]. By this method, the studied *C. psittaci* strains are classified into four main groups (I to IV), in connection with their avian hosts (parrot, duck, pigeon, turkey), the WC *C. psittaci* mammal strain being apart. Knowledge about the genotypes circulating in wildlife (e.g., sea birds and birds of prey) remains limited.

Genotyping based on *ompA* is commonly used, but resolution is rather low and since the gene is a hot spot for mutations and recombination [10,11], it may give misleading results. MLST provides higher discrimination and is robust, but requires time-consuming sequencing of seven gene targets. With the increase in number of whole genome sequences (WGS) available for *C. psittaci* strains [12,13,14,15,16], it is now possible to apply single nucleotide polymorphisms (SNPs) for the establishment of a phylogenetic tree pointing out true relationship between *C. psittaci* isolates. In our study, we aimed to develop a highly discriminative and user-friendly method where the work comprises three parts: (i) to identify clusters, based on SNP analysis, (ii) to compare those with MLST- and *omp*A-based topologies and (iii) to develop a rapid typing tool based on the identification of relevant SNPs by PCR/high-resolution melting (HRM) technology for straightforward genotyping of *C. psittaci* from isolates and clinical samples.

## 2. Materials and Methods

### 2.1. Bacterial Strains and Isolates

Strains and/or DNA samples were obtained from Anses (France), Friedrich-Loeffler-Institut (Germany), Veterinary Research Institute (Poland), University of Bordeaux (France) and Uppsala University Hospital (Sweden) (Table 1, Appendix A). Chlamydial strains were propagated in the yolk sac of chicken embryos or by cell culture as previously described [17,18]. DNA from strains or samples collected for diagnostic purposes was extracted using the QIAamp DNA Mini Kit (QIAGEN, Courtaboeuf, France). All samples included in this study tested positive for *C. psittaci* using the PCR system developed by Pantchev et al. [19].

### 2.2. Whole Genome Phylogenetic Analysis and Selection of PCR-HRM Markers

The genomic sequences of 55 available *C. psittaci* strains used for comparison are listed in Appendix A. The whole genome SNP (wgSNP) pipeline of the BioNumerics software v7.6.1 (Applied Maths, Sint-Martens-Latem, Belgium) was used in order to detect SNPs on whole genome sequences and perform cluster analyses on the resulting wgSNP matrix. The input of the wgSNP module is raw data except for the reference. Each genome file were processed with the ART-MountRainier-2016-06-05 simulation tool that generates synthetic paired-end reads with coverage of 50 [20]. These reads were aligned and mapped against the reference sequence *C. psittaci* 6BC (CP002549.1) using the BWA algorithm implemented in BioNumerics with minimum 90% of sequence identity. Strain-specific SNPs were identified using the BioNumerics wgSNP module and then filtered using the following conditions: minimum 5× coverage to call a SNP, removal of positions with at least one ambiguous base, one unreliable base or non-informative SNP and minimum inter-SNP distance of 25 bp. A phylogenetic tree built using RAxML version 8.2.9 with the GTRGAMMA model and 1000 bootstrap replicates based on the filtered SNP matrix (4143 SNPs) from BioNumerics [21]. These SNPs are distributed throughout the genome.

For the eight lineages identified (groups I to VIII), an SNP was randomly selected and PCR-HRM primers were designed using Primer3Plus software [22]. The post-real-time-PCR HRM analysis offers the possibility to detect sequence variation inside amplicons without the need for sequencing or sequence-specific probes. With high precision, the melt profile of the PCR products is determined using double-stranded DNA binding dyes and accurate fluorescence data acquisition over small temperature increments. This method enables the discrimination of amplicons differing in a single SNP, according to their melting temperature (Tm). The positions of the selected SNPs in the *C. psittaci* 6BC genome and the primer sequences used in this study are listed in Table 2 and Table 3, respectively.

### 2.3. PCR-HRM Assay

For samples from human or animal origin with a low DNA content (Cq higher than 33 with the *C. psittaci* real-time PCR), a pre-amplification step was done to increase the amount of DNA template (using the Perfecta^®^ pre-amplification kit (Quantabio) and a mix of the eight set of primers for 15 cycles).

PCR-HRM amplifications were performed on the ViiA7™ Real-Time PCR System (Life Technologies, Carlsbad, CA, USA) using the LightCycler^®^ 480 High Resolution Melting Master Mix (Roche Diagnostics, Roche, Switzerland). The reaction mixture consisted of 0.2 μM of each primer, 1 × LightCycler^®^ 480 HRM master mix and 2.5 mM MgCl2 in an 18-μL final volume. The following parameters were used: 10 min at 95 °C were followed by 40 cycles consisting of 10 s at 95 °C, 10 s at 60 °C and 20 s at 72 °C. Samples were next heated to 95 °C for 30 s, cooled down to 65 °C for 1 min and heated from 65 °C to 88 °C at a rate of 1 °C/s with 25 acquisitions/°C. HRM data were analyzed by the ViiA7™ Software (version 1.2.1). Synthetic oligonucleotide templates were PCR amplified and used as controls for each marker (dilution of 10^−7^ from a 100 µM solution) in HRM analysis.

### 2.4. ompA and MLST Typing

*omp*A sequencing using primers 3GPB and 5GPF was performed as previously described [23]. The seven housekeeping genes of the MLST method [9], namely *gat*A, *opp*A, *hfl*X, *gid*A, *eno*A, *hem*N and *fum*C, were amplified and sequenced using primers and conditions described on the *Chlamydiales* MLST website (http://pubmlst.org/chlamydiales/http://mlst.ucc.ie/, accessed on 1 February 2020). Sequencing of both DNA strands was performed by Eurofins (Reichenwalde, Germany) and numbers for alleles and sequence types (STs) were assigned in accordance with the *Chlamydiales* MLST Database.

### 2.5. In-Silico MLST Typing

Assemblies of the 55 *C. psittaci* strains were downloaded from NCBI and a script (https://github.com/tseemann/mlst, accessed on 1 February 2020) was used to scan contig files against the PubMLST *chlamydiales* scheme. The resulting table was used to build a tree in BioNumerics using the parameter “categorical values” to calculate the similarity matrix and UPGMA to reconstruct the tree.

## 3. Results and Discussion

The availability of complete genome sequences of a large panel of *C. psittaci* strains allowed the establishment of phylogenetic relationships between *C. psittaci* isolates. Using a large number of SNPs scattered through the genomes of 55 strains, the construction of a SNP-based tree led to the identification of eight distinct lineages (Figure 1), all correlating with the currently defined genotypes. Indeed, in this tree, most of the strains are distributed into three main groups: strains isolated from psittacine birds (genotype A), ducks (genotypes C and E/B), or pigeons (genotypes B and E). This clustering is consistent with the MLST clustering described by Pannekoek et al. [9], with strains of *C. psittaci* grouped mainly according to the bird groups they infect. The other five groups included the more anecdotal genotypes, such as NJ1 associated with turkeys (genotype D), VS225 (genotype F), Mat116 (previously proposed without distinct host species assignments [7]), as well as the mammalian genotypes related to the WC or M56 strains. These eight groups were named: group I_psittacine, group II_duck, group III_pigeon, group IV_turkey, group V_Mat116, group VI_M56, group VII_VS225, and group VIII_WC.

Based on this clustering, a specific SNP for each of these eight groups (named SNP1 to 8) was selected and primers designed for a specific amplification by PCR. The HRM curves obtained for the eight *C. psittaci* targeted SNPs are shown in Figure 2. All SNPs allowed a clear distinction between amplicons from the different targeted groups. Synthetic oligonucleotides corresponding to PCR amplified fragments were used as template controls for each marker, as well as DNA from an initial set of 11 reference strains of *C. psittaci* (except Mat116, not available). All these samples clustered in their intended group and all yielded amplicons producing a single melting peak, with Tm values depending on the SNP carried by the amplicon. On average, differences in Tm values of about 0.3 to 1.1 °C were observed between the paired amplicons specific of each group (Appendix A).

The developed PCR-HRM method was then applied to 118 DNA preparations from *C. psittaci* strains isolated from different avian or animal hosts. Results are summarized in Table 1. All strains isolated from psittacine birds, ducks, or pigeons clustered in their respective group, except 91-5983, isolated from a psittacine that clustered in the pigeon group (group III). In line with this result, this strain was previously typed as genotype E by PCR-restriction fragment length polymorphism (RFLP) and microarray [24], a genotype commonly associated with pigeons. A strain of unknown origin (87-1365), clustered in the psittacine group (group I_psittacine) and was previously characterized as genotype A. Most of the strains included in this study were genotyped with the microarray test and/or PCR-RFLP and/or ompA sequencing (*n* = 94/117), and identical clustering results were obtained for all of them (Table 1, Appendix A). Field strains without preliminary genotyping clustered into the group corresponding to their bird host (group I_psittacine (*n* = 13), group II_duck (*n* = 5) or group III_pigeon (*n* = 1)). It is interesting to note that the group I_psittacine also includes strains of *C. psittaci* isolated from non-avian hosts (rabbit, pig, tick, rat) indicating a certain extent of variability in host tropism of *C. psittaci* strains. Shared grazing and mixed farming may also explain the detection of a duck genotype (group II_duck) in a ruminant (C1/97) or a pigeon genotype (group III_pigeon) in a pig (01DC12).

The strain 99DC05 isolated from a horse in Germany [25] clustered in group V_Mat116 and Ful127, a strain recently isolated from a fulmar (Procellariidae) [26], clustered in group VIII_WC. The MLST sequences of these two strains are both close to the MLST ST24, corresponding to the reference strain 6BC (group I_psittacine), with only two mutations in the *hfl*X gene for Ful127 and one mutation in the *eno*A gene for 99DC05. In this case, the SNP-based typing generates a different topology than MLST analysis (Appendix A). Our study was conducted on a large set of SNPs (4143 SNPs) distributed through the genome of *C. psittaci* strains whereas only seven housekeeping genes were analyzed for MLST. It is likely that WGS analysis will help, in the near future, to refine the classification of different genotypes.

In a second step, the PCR-HRM method was applied to samples of animal or human origin. Of the 17 animal samples, seven were tissues from which strains included in the study had been isolated (six from pigeons (10-743_SC1, 10-743_SC22, 10-743_SC24, 10-743_SC28, 10-743_SC33 and 10-2168) and one from a psittacine (10-1735)). Identical clustering results were obtained. Other samples from psittacine birds (17-10114 and 17-10090), duck/waterfowl birds (15-46D/8, 15-53D/8, 15-41/8, 15-63/3) or a hooded crow (15-8D/13) gave a consistent clustering with the bird groups, confirmed for most of them by *omp*A sequencing results when available, except for 15-57D/2. Indeed, this sample came from a black-headed gull and was *omp*A-genotyped as Mat116-like [8], whereas the PCR-HRM analysis clustered it in group II_pigeon. This discordant result could be due to the well-known heterogeneity of the *omp*A gene being a hot spot for mutations and recombination events [11].

Furthermore, PCR-HRM typing confirmed that two ruminant samples (12-2090_P088 and 12-1950_M074) belonged to group VI_M56, in agreement with previous *omp*A analysis. Interestingly, these two samples were isolated from cases of cattle abortion illustrating that this mammalian genotype of *C. psittaci* can be associated with reproductive failure in cattle.

When human cases are suspected, it is important to obtain information on the source of infection to avoid the spread of the infection in animals, but also to potentially prevent further human exposure. It is often difficult to determine the origin of infection outside a workplace or a family context, with a clear and identified exposure to birds. Indeed, strains of *C. psittaci* are hosted by a variety of birds, ranging from domestic birds, to exotic and wild birds, and direct or indirect exposures can be linked to a variety of activities. The low amount of chlamydial DNA in human samples, especially when non-invasive samplings are performed (throat swabs, nasopharyngeal aspirates), is a limitation for a sequence-based genotyping that also requires time. In this study, the PCR-HRM typing scheme was applied to 20 human samples after a pre-amplification step, but only 13 were successfully typed. The limit of amplification for typing, determined from the synthetic control templates, was estimated at 5 × 10^4^ copies per µL. The seven unamplified samples were very weakly positive with Cq > 38 in the *C. psittaci* real time PCR, which explains the failure to generate amplicons for PCR-HRM. Six of the successfully amplified samples were collected from duck breeders (16-1264_MJC, 16-1264_JL1, 16-1264_JL2, 2008_A, 2009_A, and 20-3954_C054) and genotyped as group II_duck by PCR-HRM) and three other samples (16-1264_VE, 20-1105_A036, and 20-1105_A039) were genotyped as group III_pigeon by PCR-HRM, in line with the suspected infection source (dust exposure, cleaning of attics). These bird groups are among the common sources of human infection [3].

In particular, the PCR-HRM method was applied to the sample 17-5203 from a deceased patient whose infection source was not clearly established at the time of hospitalization, as the person had both raised poultry and hunted game shortly before the onset of clinical signs. PCR-HRM analysis identified the group II_duck genotype, presumably from wildlife, as all backyard birds tested negative. This result was confirmed by analysis of the *omp*A (genotype C) and MLST (ST28) results.

Interestingly, analysis of human samples recently collected in France (20-1105-A041) and in Sweden (19-5617_I078_G2662, 19-5617_I080_K12016) revealed the group V_Mat116 genotype, a genotype poorly described so far and also isolated from a horse in Germany (strain 99DC05). While the original Mat116 strain was isolated from an unspecified psittacosis outbreak in Japan (Genbank CP002744), the French and Swedish human cases are likely to have wild birds as the probable origin, since no contact with poultry or psittacine birds has been identified.

The PCR-HRM typing tool presented in this study should be considered as a scheme under development, which should be enriched with sequencing data as genomes are contributed and which may then require an update of the SNP and corresponding primer panels. Indeed, during the development work of this study, the PCR-HRM analysis performed with the initially established set of primers (not shown), did not allow affiliation of the recently isolated strain Ful127 to one of the eight determined groups and a new SNP marker for the group VIII_WC had to be implemented.

In summary, our findings show that this first set of PCR-HRM markers can be used as a new typing method providing discrimination between *C. psittaci* isolates from diverse hosts. Moreover, as additional chlamydial genome sequences become available, it will be possible to search for new SNP markers that can be used for further strain discrimination. Given the zoonotic risks associated with *C. psittaci* infection, a prompt source determination is crucial for a good sanitary management of cases and prevention of human exposure.

## Figures and Tables

**Figure 1 microorganisms-09-00625-f001:**
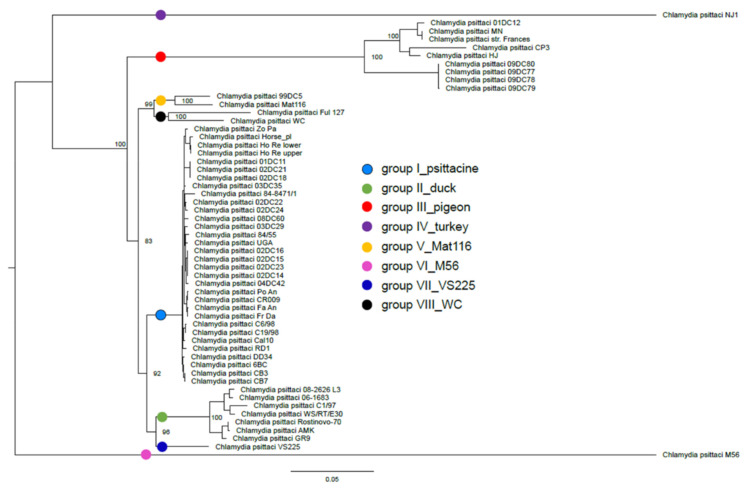
Maximum likelihood SNP-based tree determined from 55 *C. psittaci* whole genome sequences. The tree was built using RAxML version 8.2.9 with the GTRGAMMA model and 1000 bootstrap replicates. The eight distinct lineages determined in this study (group I_psittacine, group II_duck, group III_pigeon, group IV_turkey, group V_Mat116, group VI_M56, group VII_VS225, and group VIII_WC) are represented by coloured circles. Bootstrap values indicate the stability of the branches and the scale bar represents the number of substitution per site.

**Figure 2 microorganisms-09-00625-f002:**
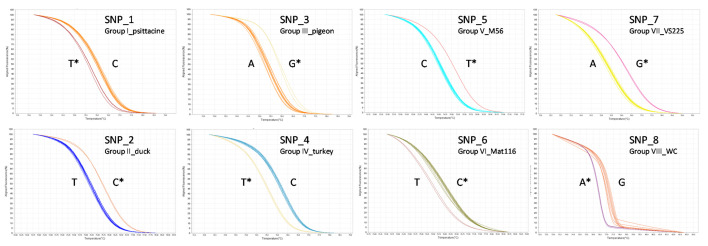
Normalized melting curves obtained with the eight SNP markers for reference *C. psittaci* strains and the respective synthetic controls. The SNP specific to the targeted group is noted by an *.

**Table 1 microorganisms-09-00625-t001:** PCR-high-resolution melting (HRM) clustering of reference strains, field strains and clinical samples. The host origin and available genotype and/or MLST results are shown in Appendix A.

HRM Group	Strain/Sample	ID
**I_Psittacine**	reference strains	6BC-04DC45, Loth, VS1
	field strains with typing data	84-6461, 84-8471/1, 84-9462, 85-1173, 85-12098, 86-0191, 86-10703, 86-14356, 86-3389, 87-13654, 88-2014, 88-5558, 88-5821, 88-8795, 89-2930, 90-0057, 90-0475, 90-10445, 90-11404, 90-12937, 90-4862, 91-14273, 91-5189, 91-6047, 95-1334, 97-5075, 97-6475, 98-7627, 99-0182, 99-0923, 99-8157/1, 00-0151, 00-0476, 00-1176, 00-1268, 00-1750, 05-0949, 05-4098, 06-0372, 06-0852, C6/98, 01DC11, 03DC29, 03DC35, 04DC42, 04DC46
	field strains without typing data	96-6274, 96-12328, 96-12742, 97-822, 97-9244, 99-0313, 99-1394, 99-8888, 00-4462, 03-3227, 04-2668, 10-0485, 10-1735
	clinical samples	10-1735 *, 17-10114, 17-10090
**II_Duck**	reference strain	GR9
	field strains with typing data	94-2306, 05-4325, 05-4461, 06-859, 06-871, 06-881, 06-889, 06-1683, 10-1398/28, 10-1400, C1/97, 07-1391, 08-2626_L3, 08-2626_L4
	field strains without typing data	04-5006, 05-553/17, 07-2962, 08-2850, 10-1393
	clinical samples	15-46D/8, 15-53D/8, 15-41/8, 15-63/3, 16-1264_MJC, 16-1264_JL1, 16-1264_JL2, 17-5203, 2008_A, 2009_A, 20-3954_C054
**III_Pigeon**	reference strains	CP3, Cal10, MN
	field strains with typing data	90-12617, 91-6568, 91-12516, 89-13210, 91-5983, 05-4036, 09-295_JF5, 09-295_JF8, 09-295_J9, 09-295_MB32, 09-295_MB33, 09-336_FA30634, 09-487_T13, 09-489_LP5, 09-489_LP7, 09-489_Mon5, 09-489_Mon13, 09-496_FA32303, 09-496_FA32311, 09-544_Van14, 09-589_S10, 09-928, 10-743_SC1, 10-881_SC22, 10-743_SC24, 10-743_SC28, 10-743_SC33, 10-743_SC42, 10-881_SC42, 10-883_EL27, 10-1048_Bat16, 01DC12, 03DC32, 09DC75, 11DC94
	field strain without typing data	10-2168
	clinical samples	15-8D/13, 15-57D/2, 10-743_SC1 *, 10-743_SC22 *, 10-743_SC24 *, 10-743_SC28 *, 10-743_SC33 *, 10-2168 *, 16-1264_VE, 20-1105_A036, 20-1105_A039
**IV_Turkey**	reference strains	NJ1, TT3
**V_Mat116**	field strain with typing data	99DC05
	clinical samples	20-1105_A041, 19-5617_I078_G2662, 19-5617_I080_K12016
**VI_M56**	reference strain	M56-07DC57
	field strain with typing data	16DC111
	clinical samples	12-2090_P088, 12-1950_M074
**VII_VS225**	reference strain	VS225
**VIII_WC**	reference strain	WC-07DC58
	field strain with typing data	Ful127

* linked to a field strain.

**Table 2 microorganisms-09-00625-t002:** List of selected single-nucleotide polymorphisms (SNPs) used for this study. SNPs specific to each group are in bold.

		SNP for Each Group										
SNP No.	Associated Group	Group I	Group II	Group III	Group IV	Group V	Group VI	Group VII	Group VIII	Position on 6BC	Gene	Locus Tag
		(Psittacine)	(Duck)	(Pigeon)	(Turkey)	(M56)	(Mat116)	(VS225)	(WC)			
**1**	**Group I_Psittacine**	T	C	C	C	C	C	C	C	126074	lipoate-protein ligase family protein	CPSIT_RS00640
**2**	**Group II_Duck**	T	C	T	T	T	T	T	T	39653	CesT family type III secretion system chaperone	CPSIT_RS00155
**3**	**Group III_Pigeon**	A	A	G	A	A	A	A	A	1038463	cation-translocating P-type ATPas	CPSIT_RS04480
**4**	**Group IV_Turkey**	C	C	C	T	C	C	C	C	1352	Na(+)-translocating NADH-quinone reductase subunit A	CPSIT_RS00010
**5**	**Group V_M56**	C	C	C	C	T	C	C	C	961	hemB	CPSIT_RS00005
**6**	**Group VI_Mat116**	T	T	T	T	T	C	T	T	85629	anti-sigma regulatory factor	CPSIT_RS00375
**7**	**Group VII_VS225**	A	A	A	A	A	A	G	A	13908	exodeoxyribonuclease V subunit gamma	CPSIT_RS00045
**8**	**Group VIII_WC**	G	G	G	G	G	G	G	A	722388	YqgE/AlgH family protein	CPSIT_RS03140

**Table 3 microorganisms-09-00625-t003:** PCR primers used for the PCR-HRM analysis.

SNP No.	Associated Group	Forward Primer (5′-3′)	Reverse Primer (5′-3′)	Amplicon Size (bp)
**1**	Group I_Psittacine	gacccaacgagatttctgga	cccaaagacatttgccttaca	94
**2**	Group II_Duck	gcgatctcgtcaagatacgtg	ttggtatccgaagaaggaggt	94
**3**	Group III_Pigeon	cttctttcttgcaggaactccag	atccgaaagctgctgacgtc	101
**4**	Group IV_Turkey	aagaaccctaacatgcacgc	ggcgatgaaaatccctgttgt	81
**5**	Group V_M56	tgatgtgttgcatcgagtga	ccactgacttgataggctgct	63
**6**	Group VI_Mat116	cgcttcttggtatgcataggag	agaacaactcaaaacattcccaa	100
**7**	Group VII_VS225	aagggagtcagaagaagagaaaa	actaatgctacgagtaaccacg	63
**8**	Group VIII_WC	tgaacaggaatgcaaaagca	tgggtttagaaatagctgacga	119

## Data Availability

Detailed data are available upon request.

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
