# Peer review of "A New SNP-Based Genotyping Method for C. psittaci: Application to Field Samples for Quick Identification"

_microorganisms, 2021, doi:10.3390/microorganisms9030625_

Round 1
Reviewer 1 Report
This manuscript describes analysis of single nucleotide polymorphisms in C. psittaci genomes and development of a PCR /high resolution melting assay for identification of eight C. psittaci lineages.
The assay was validated using reference strains and field isolates. It was then applied to studying 28 samples from animals and humans. Results obtained were in agreement with earlier, more tedious typing methods (ompA analysis and MLST).
This manuscript describes a novel, rather rapid method for typing C. psittaci, a most likely neglected pathogen among animals and humans. The manuscript is well-written.
I have some minor comment/question:
The first sentence in the introduction states that C. psittaci is commonly isolated from a wide range of birds. I would like to see actual figures (how often) and is this a phenomenon observed widely (which countries? Continents?).
Were the selected amplicon sizes optimal for showing the difference in signal at a given temperature between the sequences that differed at only one nucleotide position?
A pre-amplification step was included for biological samples.
What were “biological” samples (samples of animal or human origin?)? Please define.
Please describe in detail the preamplification step (no of cycles, Exo I treatment?).
Did you assess how much this actually improved the yield?
If I understood correctly, all animal samples, but 13/20 human samples could be typed.
Author Response
please see the attacment

Reviewer 2 Report
Thank you for the opportunity to review this manuscript, the authors have outlined a potentially useful method for typing Chlamydia psittaci strains more rapidly than using currently available ompA or MLST sequencing approaches. I have some general comments, and section specific comments, broken into what I consider major (should be addressed for publication) and minor (suggestions and possibly some personal preferences rather than necessary changes) that I believe will improve the the manuscript and clarify aspects of the study. I believe with these comments addressed the manuscript will be suitable for publication in Microorganisms.
General comments/suggestions
Minor
- There are several sections in the manuscript that say ‘as described by [#]’, it may be a personal preference (and not the journals?) but ‘as described by Smith et al [#]' is slightly easier to read.
- The discussion mentions recombination as a potential reason for discordant results, but it’s unclear whether the SNP selection method excludes regions of potential recombination when coming up with phylogenetically relevant regions. The 12 bp distance requirement between SNPs may assist with this using the assumption that recombination regions have a higher SNP abundance, but it should either be taken into consideration or discussed in the methods or results (respectively).
Introduction
Minor
- A small section on why SNP/HRM methods are more suitable than full genome sequencing would perhaps add value to the introduction section (why hasn’t it been done before, why is better than ompA/MLST, why is it faster/more reliable than aiming for a genome at a time). I think this would fit as part of the final paragraph of just prior to it
Methods
Major
- I’d suggest the authors use a maximum likelihood approach with an incorporated evolutionary model and ascertainment bias correction for SNPs (potentially using a tool such as IQTree) to produce their main phylogenetic tree unless a suitable reason exists for using maximum parsimony? This may be deemed a personal preference, but they cite similarities in phylogenetic clustering with Pannekoek’s MLST trees, and that paper used a likelihood rather than parsimony approach, so comparing two trees produced using different methods is likely to lead to misinterpretations.
- Line 76-77 - A supplementary table of the 54 C.psittaci strains used, with the appropriate NCBI accession numbers, should be included instead of the link to the NCBI website, which is dynamic and assembly versions may change or new assemblies added. This table would save those wanting to use this paper (or build on it in the future, as suggested by the authors) many headaches.
- Some information about how the biological samples were collected is needed (e.g. were they diagnostic samples or were they collected for the project, under which scenario animal ethics approval should be outlined).
- Line 88 – Primer3Plus should be cited (Andreas Untergasser, Harm Nijveen, Xiangyu Rao, Ton Bisseling, René Geurts, and Jack A.M. Leunissen: Primer3Plus, an enhanced web interface to Primer3 Nucleic Acids Research 2007 35: W71-W74; doi:10.1093/nar/gkm306)
- Table 2 – Either in the table or in the results, there should be some information as to the location of the SNPs from a gene context. Not just the position in the chromosome, but also which genes the SNP occurs in and perhaps how variable that gene is across the different genomes assessed.
Minor
- A figure of the circularised 6BC chromosome with the ~4K SNPs highlighted (using BRIG or some similar visualisation software) would be a valuable visual aid to the reader to identify how the SNPs are spread throughout the genome. Alternating colours based on whether they’re intergenic SNPs or not, or non-synonymous/synonymous (and perhaps more/less prone to selection) would also be helpful.
- Table 1 is quite difficult to read in manuscript/printed form. This may not be an issue with online reading, but perhaps a summarised version of this data would be more suitable, with the full table available as supplementary material?
- The structure of section 2.2 is slightly confusing, it explains that the input of the wgSNP module is going to be before the reader knows what it’s doing? A brief introductory sentence would clarify what this tool is being used for.
- Line 82 - I’m not totally sure, but should ‘90% parameter identity’ be inverted to ‘90% identity parameter’?
- Lines 91-94 – this information is more suitable to an introduction than the methods section
- Line 109 – Perhaps a very short few sentences describing fragment sizes obtained for these genes would be useful. I’m only trying to avoid a reader or interested clinical diagnostician having to look up other papers for information to compare the outcomes with this paper. To be clear, I’m not looking for cycling conditions or primer mixes.
Results
Major
- Estimated limits of detection should be included for the qPCRs being undertaken (using diluted synthetic controls potentially prior) for future users of this method to refer to (if one amplicon can be detected at 100 copies, but another can only be detected at 10000 copies, this is important information)
- The bootstrap values don’t appear to be correct. I believe these should be between 0-1 (or 0 – 100). There are branch values provided >1000, they appear to represent substitution distances or similar rather than bootstrap values. These need to be changed for readers (or reviewers) to correctly view/interpret the grouping/branching. I’d also encourage a higher number of bootstraps (~1000) considering the number of sites in the tree
- It is mentioned that 7/20 human samples were not successfully typed but there is no information as to why. Were they unable to be detected with these PCRs (one or more amplicons?), poor melt curves, or was it due to low quantities of DNA in the sample? This is important to highlight if the aim of the manuscript is to describe the validity of a method.
Minor
- Some additional information for the phylogenetic trees should be included in their figure legends. It’s unclear whether the figure 2 tree is the same tree referred to in the methods (which used maximum parsimony). And no information is provided for the method of producing the Supplementary Figure 1 tree.
- Line 136-137 – ‘Data not shown’ should at least be supplementary data in an online journal
- The term “Fulmar” should perhaps be explained for a non-avian audience with either a scientific name or reference to the family.
- Lines 195 – 199 – This paragraph states that the detected Mat166 genotype samples from humans are likely from wild bird origin, but provide no discussion or evidence as to why this conclusion has been reached. This should be expanded on (or removed).
